# Computational modelling of hip resurfacing arthroplasty investigating the effect of femoral version on hip biomechanics

Jonathan Bourget-Murray[1]*, Ashish Taneja[1], Sadegh Naserkhaki[2], Marwan El-Rich[3], Samer Adeeb[3], James Powell[1], Kelly Johnston[1]

1 Department of Surgery, Section of Orthopaedic Surgery, University of Calgary, Calgary, Canada, 2 Department of Biomedical Engineering, Science and Research Branch, Islamic Azad University, Tehran, Iran, 3 Department of Civil and Environmental Engineering University of Alberta, Edmonton, Alberta, Canada

* jbourgetmurray@gmail.com

**Data Availability Statement:** All relevant data are within the paper.

## Abstract

### Aim

How reduced femoral neck anteversion alters the distribution of pressure and contact area in Hip Resurfacing Arthroplasty (HRA) remains unclear. The purpose of this study was to quantitatively describe the biomechanical implication of different femoral neck version angles on HRA using a finite element analysis.

### Materials and methods

A total of sixty models were constructed to assess the effect of different femoral neck version angles on three different functional loads: 0° of hip flexion, 45° of hip flexion, and 90° of hip flexion. Femoral version was varied between 30° of anteversion to 30° of retroversion. All models were tested with the acetabular cup in four different positions: (1) 40°/15° (inclination/version), (2) 40°/25°, (3) 50°/15°, and (4) 50°/25°. Differences in range of motion due to presence of impingement, joint contact pressure, and joint contact area with different femoral versions and acetabular cup positions were calculated.

### Results

Impingement was found to be most significant with the femur in 30° of retroversion, regardless of acetabular cup position. Anterior hip impingement occurred earlier during hip flexion as the femur was progressively retroverted. Impingement was reduced in all models by increasing acetabular cup inclination and anteversion, yet this consequentially led to higher contact pressures. At 90° of hip flexion, contact pressures and contact areas were inversely related and showed most notable change with 30° of femoral retroversion. In this model, the contact area migrated towards the anterior implant-bone interface along the femoral neck.

### Conclusion

Femoral retroversion in HRA influences impingement and increases joint contact pressure most when the hip is loaded in flexion. Increasing acetabular inclination decreases the area

**Funding:** The author(s) received no specific funding for this work.

**Competing interests:** The authors have declared that no competing interests exist.

of impingement but doing so causes a reciprocal increase in joint contact pressure. It may be advisable to study femoral neck version pre-operatively to better choose hip resurfacing arthroplasty candidates.

## Introduction

There is evidence that Hip Resurfacing Arthroplasty (HRA) offers better function in daily life, higher activity, and better general physical health compared to Total Hip Arthroplasty (THA) in young active males [1]. This patient population have historically experience decreased overall satisfaction and higher complication rates when THA is performed [2]. Advocates of HRA suggest fewer dislocations, decrease proximal femoral stress shielding, and improved restoration of a more anatomic hip that allow for greater motion [3–5]. In addition, HRA provides potential for return to high-level activities [6]. Equally important, the conservative resection of femoral bone preserves bone stock, thus facilitating future revision surgery [4,7]. Although the use of these implants has substantially dropped due to concerns of adverse local reactions from metal-ion debris and the possible increased risk of femoral neck fractures, the Birmingham Hip Resurfacing system has remained in use and several centers continue to report excellent long-term clinical and functional outcomes [8–14].

In light of these adverse events, several studies have identified risk factors attributed to early failure, which include: implant design, component position and size, female sex, patient age, and surgeon inexperience [8,12,15–17]. As a means to investigate the association between femoral implant malposition on damage formation of the femur, a recent finite element analysis was developed. This model was developed from a 47-year old patient's computed tomography (CT) image and the loading simulation was that of normal walking condition. The analysis showed that the model experienced the most damage when the femoral head implant was in varus position (>130˚), but was reduced significantly when the implant was placed in valgus position (<130˚) [16]. On the other hand, Ramos et al. developed experimental and numerical models to analyze whether the positioning of the resurfacing head implant is important in the distribution of bone strains and in the risk of fracture of the femur [17]. They found that valgus position reduces strain distribution in the medial aspect of the femur and brings about a lower shear stress, thus reducing the risk of femoral neck fracture. However, despite appropriate patient selection and precise implant positioning, there continues to be a subset of patients who experience early failure following HRA. Structural abnormalities of the hip may serve as an explanation for some of these failures.

Patients who present with end-stage arthritis of the hip can have contributing underlying structural abnormalities, such as acetabular dysplasia or femoroacetabular impingement (FAI). Cam and pincer impingements are well described but reduced femoral neck anteversion is less obvious and can be a major contributor to FAI and ultimately, hip arthritis [18]. While the biomechanical implications of acetabular component malposition has been extensively studied, that of femoral version remain unclear. This structural abnormality may be responsible for ongoing impingement and may increase the risk of femoral neck fracture. Perhaps some of the unique complications associated with HRA can be explained by understanding how reduced femoral version (femoral retroversion in extreme cases) impacts force transmission across the hip joint.

The purpose of this study was to quantitatively describe the biomechanical implication of varying femoral version (including femoral retroversion) on HRA with respect to

impingement and force transmission across the joint. The secondary goal was to quantify the impact that different acetabular cup positions have on impingement and joint reaction forces. We hypothesize that pre-existing femoral retroversion leads to abnormal joint mechanics and could therefore be a risk factor for premature failure following HRA.

## Materials and methods

Three-dimensional (3D) geometry of a right cadaveric donor hemipelvis and proximal femur was reconstructed from a CT-scan of 0.5 mm slice thickness to create a finite element (FE) model of the right hip joint. Segmentation was performed using the medical image processing software Mimics (Materialise, Belgium). All necessary measurements were performed using Geomagic (3D Systems, USA). The native femoral head and neck size were 54 mm and 42 mm, respectively. The native femoral neck-shaft angle was 125˚, and the femoral neck version was 20˚. A virtual model of a Birmingham Hip Resurfacing (BHR, Smith & Nephew Orthopaedics Ltd, Warwick, UK) was performed featuring a 54 mm femoral head and a 60 mm acetabular shell (Fig 1). The acetabulum was initially positioned in the standard position: 40˚ of inclination and 15˚ of version [19]. The femoral component was positioned, as per company standards, in slight valgus (-5˚). The geometry of the femoral head was shaped to simulate the cylindrical and chamfer reamings performed during the usual surgical operation with a layer of cement between the femoral head and metal covering [20,21]. The thickness of the cement mantle was 0.5 mm and was simulated by brick solid elements.

The FE mesh was generated using Hypermesh (Altair, USA). Specific characteristics of different model parts and necessities of the model (along with the research objectives) were

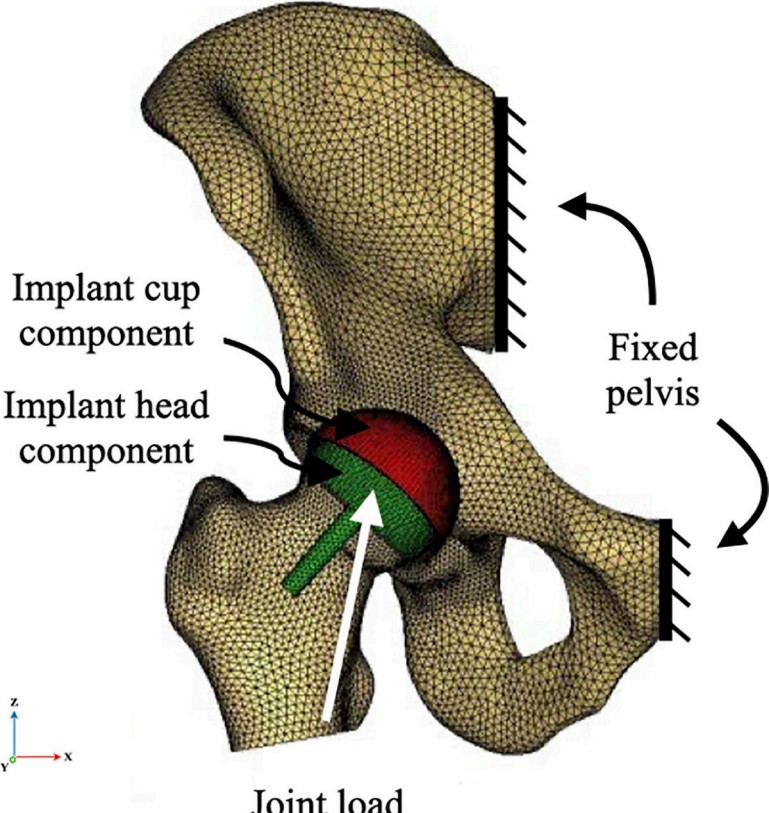

**Fig 1. Finite element (FE) model of the hip resurfacing arthroplasty (HRA).**

**Table 1. Material properties of the FE model.**

| Components | Element Type | Young modulus *E* (GPa) | Poisson Ratio |
|---|---|---|---|
| Cortical bone* (pelvis/femur) | Shell (thickness: 1.5mm) | 20 | 0.3 |
| Cancellous bone* (pelvis/femur) | Solid | 1 | 0.3 |
| Implant§ (head/cup) | Solid | 200 | 0.3 |
| Cement§ | Solid | 2 | 0.19 |

*Adopted from: Chegini S, Beck M, Ferguson SJ. The effects of impingement and dysplasia on stress distributions in the hip joint during sitting and walking: A finite element analysis. J Orthop Res. 2009:27(2):195–201. [22].

§Adopted from: Sakagoshi D. Kabata T. Umemoto Y, et al. A mechanical analysis of femoral resurfacing implantation for osteonecrosis of the femoral head. J Arthroplasty 2010;25(8):1282–1289. [25].

determinant in assigning the type of elements and their material properties. The cortical bone was meshed with 3-node shell elements with a uniform thickness of 1.5 mm, as previously described by Chegini et al, and filled with 4-node tetrahedral elements to represent cancellous bone [22]. The BHR implant and cement were meshed with 8-node hexahedral elements. In order to mesh the relatively thin implant (mesh it with several layers) and to manage the aspect ratio of the elements, we had to use very small size elements. The very fine mesh of this model made the mesh size effects negligible. There was no slippage between the femur, cement, and implant [23]. The cup was also secured to the acetabulum to restrict slippage. A surface-to-surface contact was considered between the outer surface of the head and inner surface of the cup with 80 μm of clearance [21]. This clearance was determined by property of contact element. The contact was defined by pairing two surfaces. When they got close to each other, at a defined distance of 80 μm, contact was detected. This was the initial distance that contact was initiated due to presence of fluid between the femoral head and acetabular cup. In addition, a frictional coefficient of 0.15 was considered between surfaces [24]. The cortical and cancellous bone were considered homogeneous (uniformly distributed density) with isotropic linear elastic material behaviour (Table 1). The polymethyl methacrylate (PMMA) bone cement mantle and cobalt-chrome implant were modelled as isotropic linear elastic materials.

The model did not account for movement through the sacroiliac joint or the pubic symphysis. These joints were fully constrained for the purpose of this study. The femur on the other hand, was kept free to move during loading. A concentrated load was applied to a reference point defined at the center of the femoral head (Fig 1). The applied load was adopted from *in-vivo* data of the peak hip contact force for three different loading conditions (Table 2): single

**Table 2. Loading conditions§.**

| Hip Position | Percentage of body weight* | Joint force (N) | | |
|---|---|---|---|---|
| | | $F_x$† | $F_y$† | $F_z$† |
| 0° Hip flexion | 238% | 443 | 266 | 1,920 |
| 45° Hip flexion | 251% | 492 | 510 | 1,974 |
| 90° Hip flexion | 156% | 359 | 6 | 1,253 |

§The applied load was adopted from *in-vivo* data of the hip contact force for three different loading conditions: Walking with a velocity of 1.09 m/s (0° of hip flexion), climbing (45° of hip flexion), and sitting (90° of hip flexion). Adopted from in-vivo data by Bergmann G, Deuretzbacher G, Heller M, et al. Hip contact forces and gait patterns from routine activities. *J Biomechanics* 2001;34(7):859–871. [26].

*Reference weight: 836N.

†Cartesian coordinate system with X, Y, and Z, in lateral-medial, anterior-posterior and axial directions, respectively.

leg stance (0° of hip flexion) during walking with a velocity of 1.09 m/s, climbing (45° of hip flexion), and sitting (90° of hip flexion) [13]. Center of rotation (COR) was determined by fitting a sphere over the femoral head.

To address our research question, a total of 60 FE models were constructed in order to assess the effect of varying femoral versions on the different functional loads. All the analyses were performed on the same FE model which was reconstructed from the anatomic geometry of one cadaveric hemipelvis and proximal femur. The different variations in femoral version and acetabular cup inclination was taken into account by manual alterations. The proximal femur version was varied by rotating the femur in the axial plane around the COR between 30° of anteversion to 30° of retroversion. The different femoral versions studied were (1) 30° of anteversion, (2) 15° of anteversion, (3) 0° of version, (4) 15° of retroversion, and (5) 30° of retroversion (reported from now on as: AV30, AV15, RV0, RV15 and RV30, respectively). The neck-shaft angle was maintained at 125°. All models were tested with the acetabular cup in four different positions: (1) 40°/15° (inclination/version), (2) 40°/25°, (3) 50°/15°, and (4) 50°/25°. Differences in contact pressure (MPa), contact area (mm$^2$), and impingement area (mm$^2$) were measured for all the different models.

Two different approaches were used to investigate for impingement area and contact pressure. Firstly, geometrical analyses (using Hypermesh, Altair, USA) were performed to detect and quantify the joint impingement. This included virtually simulating hip flexion whereby the femur was flexed from 0° to 90° in the sagittal plane. By way of this simulation, the degree at which impingement occurred was determined. In addition, the areas on the acetabulum and femur where the impingement occurred was captured. Secondly, in order to calculate the contact pressure and contact distribution across the joint, nonlinear stress analyses were performed using the FE solver Abaqus (Dassault Systems Simulia Corp., USA). The contact was simulated across the inner acetabular cup surface only.

## Results

### Impingement analysis

Progressive femoral retroversion was found to cause a gradual increase in the impingement area. In addition, impingement occurred earlier in the flexion ROM with progressive femoral retroversion. In the most anteverted model (AV30), impingement occurred at 70° of hip flexion, whereas impingement occurred at 30° hip flexion with 30° of retroversion (RV30).

Interestingly, the area of impingement could be reduced by either increasing acetabular cup inclination (from 40° to 50°) and/or version (from 15° to 25°). This was appreciated regardless of femoral version (Fig 2). A 10° increase in both acetabular cup inclination and version had a cumulative effect and provided the most clearance during hip flexion, thus reducing the impingement area.

### Contact pressure analysis

Negligible differences in contact pressures were seen in 0° or 45° of hip flexion across all femoral version models. However, there was a significant increase in contact pressure with progressive femoral retroversion when the hip was loaded at 90° of hip flexion (Fig 3). This trend was appreciated regardless of acetabular cup position. The RV30 model at 90° of hip flexion demonstrated the largest increase in contact pressure (18% increase) when the acetabular cup was set to 50° of inclination and 15° of version (6.3 MPa).

### Contact area analysis

The distribution of contact area was assessed across all models (Fig 4). In 0° of hip flexion, the contact area was distributed over the superior and medial areas of both the acetabular cup and

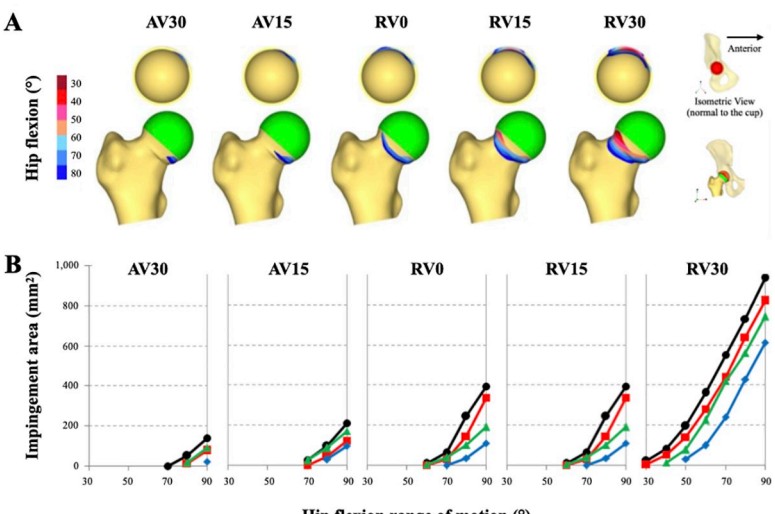

**Fig 2. Impingement area at the anteroinferior femoral neck in relation to different femoral versions and degrees of hip flexion.** (A) Schematic design showing the change in the impingement area with progressive femoral retroversion (this example accounts for an acetabular cup in 40˚ of inclination and 15˚ of version). (B) Impact of different acetabular cup positions on impingement area: 40˚ of inclination and 15˚ of version (black line); 40˚ of inclination and 25˚ of version (red line); 50˚ of inclination and 15˚ of version (green line); 50˚ of inclination and 25˚ of version (blue line).

femoral head. Femoral version had negligible impact on the contact area across the joint in this condition. In 45˚ of hip flexion, the contact area progressively migrated more posterosuperior. In 90˚ of hip flexion, the contact area tended to concentrate more towards the anterior femoral bone-implant interface. With 30˚ of femoral retroversion the contact area was maximally concentrated and contact pressures were at their highest.

The effect of different acetabular cup positions on contact area was also examined (Fig 5). When loading the hip in 0˚ or 45˚ of flexion, contact area was reduced by 2–10% in all models where the acetabular cup inclination had been increased to 50˚. Cup version was not found to have much influence on contact area. No difference was observed with different femoral versions in these conditions. Femoral version had a more noticeable influence on contact area when the hip was loaded in 90˚ of hip flexion.

Comparing the findings from Figs 3 and 5, there is an obvious inverse correlation between the smaller contact area created when the hip is loaded in 90˚ of flexion and the femur is 30˚ retroverted. This condition also was found to create the highest pressures across the HRA.

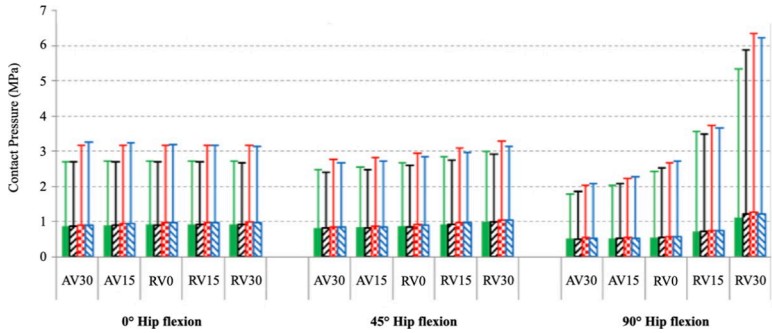

**Fig 3. Contact pressure between the acetabular cup and head in different loading conditions and acetabular cup positions.** Acetabular cup is position with 40˚ of inclination and 15˚ of version (green line); 40˚ of inclination and 25˚ of version (black line); 50˚ of inclination and 15˚ of version (red line); 50˚ of inclination and 25˚ of version (blue line).

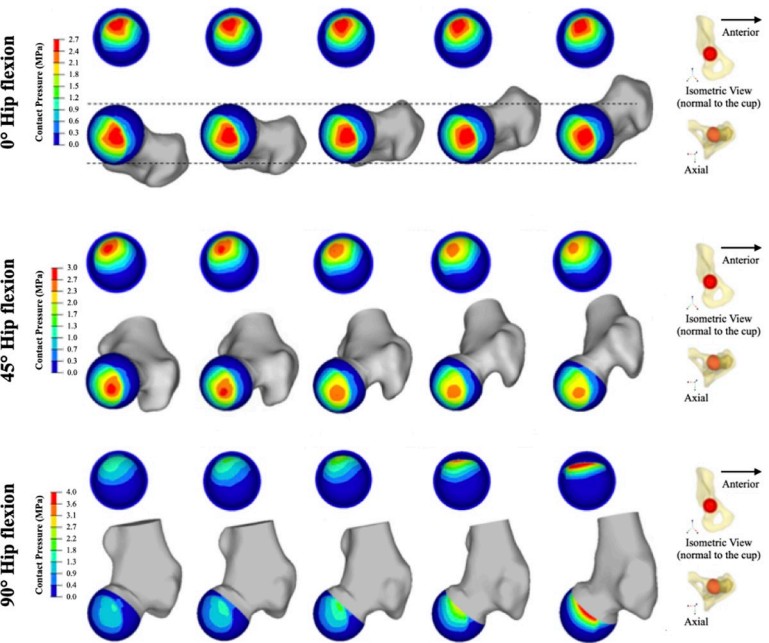

**Fig 4. Illustration of the contact area across the acetabulum and head in different loading conditions.** Contact area across the HRA when the joint is loaded in 0˚, 45˚, and 90˚ of hip flexion is influenced by femoral version (this example accounts for an acetabular cup in 40˚ of inclination and 15˚ of version).

## Discussion

Structural abnormalities of the proximal femur, particularly femoral retroversion, can contribute to the development of end-stage hip arthritis [18]. Pre-arthritic patients with femoral retroversion have profound loss of internal rotation at the hip in 90˚ of flexion and become symptomatic when the anterior femoral neck impinges on the anterior rim of the acetabulum. In light of this, femoral neck retroversion may exist in patients who receive a HRA, going unnoticed on initial evaluation because of stiffness attributed to the arthritic condition. Although component position has been identified as a risk factor for early failure following hip resurfacing, the impact of femoral version on HRA biomechanics remain elusive. Reduced femoral version (femoral retroversion in extreme cases) may be responsible for ongoing impingement and increase the risk of femoral neck fracture following surgery. Perhaps some of the unique

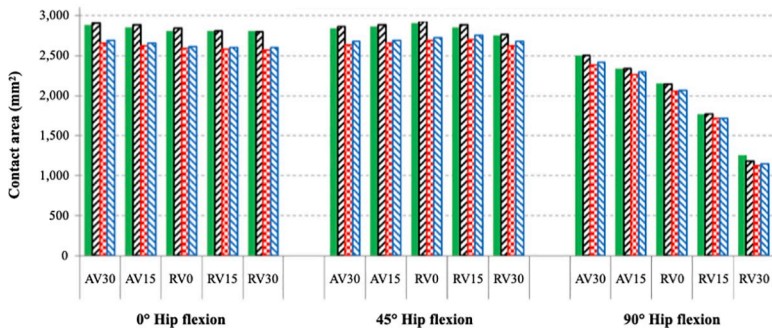

**Fig 5. Contact area between the acetabular cup and head in different loading conditions and acetabular cup positions.** Acetabular cup is position with 40˚ of inclination and 15˚ of version (green line); 40˚ of inclination and 25˚ of version (black line); 50˚ of inclination and 15˚ of version (red line); 50˚ of inclination and 25˚ of version (blue line).

complications of HRA can be explained by understanding how reduced femoral version impacts force transmission across the hip joint. To the best of our knowledge, this has never been reported. Our goal was to quantitatively describe the differences in force transmission across a hip joint with a resurfacing arthroplasty, and further understand the biomechanical implications of different femoral versions and different acetabular cup orientations on these forces.

Abnormal femoral version often alters motion of the hip and is associated with both intra-articular or extra-articular impingement. This computational analysis has shown that impinge-ment significantly increases with progressive femoral retroversion, regardless of acetabular cup position. In addition, impingement occurred earlier in hip flexion ROM as the proximal femur was progressively retroverted. This was not observed then the hip was loaded in lower amounts of flexion (0˚ or 45˚ of flexion). Only in full hip flexion (90˚) were contact areas impacted by progressive femoral retroversion. These became progressively more concentrated and subsequently contact pressures were measured to be much higher. This was most obvious when the femur was 30˚ retroverted.

Component malposition plays an important role in early failure of HRA. In fact, acetabular component malposition has been extensively studied. A poorly positioned acetabular cup can cause increased joint reactive forces and edge loading which can consequently lead to signifi-cant increase in implant wear—increasing metal-ion debris and possibly local tissue reactions [7,27]. Amstutz et al. suggested that increased acetabular inclination and anteversion was asso-ciated with increased wear, and thus recommended an optimal acetabular cup position of 42 +/- 10˚ of inclination and 15 +/- 10˚ of anteversion [7]. Any cup inclination above 50–55˚ has previously been associated with increased wear rates and increases in whole blood concentra-tions of cobalt and chromium ions after HRA [28,29]. The incidence of pseudotumours when the acetabular cup is positioned within optimal parameters (45˚ of inclination and 20˚ of ante-version) is four times lower ($p = 0.007$) than when outside these parameters [30]. Hart et al. sought to understand the variation in wear rates in a large series of 276 (138 femoral head and acetabular cup couples) retrieved metal-on-metal (MoM) hip arthroplasty components [15]. Through a multivariate analysis they found that edge-loading was the most important predic-tor of wear rate and occurred in two-thirds (88/138, 64%) of patients with failed MoM hip replacements. In those that edge-loading occurred, the likely factors involved are related to the patient (e.g., activity type), the surgery (e.g., cup orientation), and the manufacturing (e.g., cup coverage arc) [15].

While the consequences of acetabular cup malposition are well understood, recent studies have investigated the impact of femoral implant malposition on risk of failure [16,17]. Izmin et al. conducted a FE analysis to determine the bone damage that would result from different femoral implant malpositions relative to a natural femoral neck angle of 130˚: varus (> 130˚) and valgus (<130˚) [16]. The simulation performed in the study represented physiological loading of a human. Their findings suggest that an implant in valgus position reduces the stress distribution and damage formation across the femur, thus reducing the potential for fracture. These finding are in keeping with those of other studies [17,31]. However, despite ongoing research in this field, little is known about the effect of abnormal femoral version on hip impingement and contact stresses following HRA.

To quantify the effect of femoral version on native hip contact stress, Meyer et al. used five cadaveric pelvis specimens which were mechanically tested in a heel-strike position [32]. Pres-sure measurements were recorded by the Tekscan sensor with the femur oriented in 0˚, 15˚, and 30˚ of anteversion. While they did not report any difference in average peak contact stresses between different femoral versions (anteverted: 4.59 MPa; normal: 4.66 MPa; retro-verted: 4.59 MPa), they did not explore retroverted positions. While our study expands on

current literature by quantitatively describing the biomechanical implication of different femoral versions in different physiological loading conditions, these findings are similar to what we report in these respective loading conditions. In addition, this study shows that femoral retroversion imparts important changes to the biomechanics of HRA, especially when the hip is loaded in 90° of flexion. With progressive femoral retroversion, contact pressure becomes concentrated at the anterolateral femoral bone-implant interface. This may contribute to accelerated wear and possibly to femoral neck fractures. Satpathy et al. reported increased contact pressures in native hips with femoral retroversion when the hip is flexed to 90° [33]. The wear rate of HRA has previously been shown to be directly related to changes in contact pressure between the metal components [22,33]. These findings are consistent with those reported in this study. Thus, higher contact pressures across the HRA in 90° of hip flexion are accentuated with femoral retroversion, and therefore can be a potential cause of accelerated wear rate. Distribution of the contact pressure also becomes critical as the location of the maximum contact pressure appears to move toward the head-neck junction. This focal loading across a smaller contact area may explain the increased risk of a femoral neck fracture following HRA. In addition, we have shown that acetabular inclination can be increased to 50° to compensate for the increase in impingement created with progressive femoral neck retroversion. However, doing so may cause further increase in contact pressure. Ultimately, successful HRA needs to balance hip biomechanics when the hip is loaded in extension and in flexion.

Prior to the onset of arthritis, patients with reduced femoral version or femoral retroversion significant femoral neck retroversion often present with impingement symptoms and can be identified by a careful clinical exam showing excessive external rotation of the hip and profoundly reduced internal rotation when the hip ROM is passively examined at 90° of hip flexion. These same patients often present with outward foot progression angles. Once the hip is severely arthritic and stiff, the above findings may not be as easily appreciated. The only way to accurately determine the femoral neck version is with a full length CT/MRI scan of the femur, incorporating the knee to correctly measure femoral version. Given the importance of femoral neck version on hip biomechanics, perhaps this parameter should be determined preoperatively prior to consideration for HRA. Patients with significantly reduced femoral neck version, or perhaps some degree of femoral retroversion, may be better served with a THA where more normal femoral version can be restored.

Our study is not without limitations. All the models were created from one base cadaveric hip. It is obvious that the hip joint of individuals may differ substantially and will have individual specific motion and loading patterns. However, in the lack of individual specific loading, it is not uncommon to apply an average in-vivo loading to all models [32–34]. The loads placed when the hip was flexed to 90° was modelled to a static position (i.e. sitting in a chair). Much higher loads are expected to go through the joint during activities where the flexed hip is being used to support the body. Therefore, the magnitude of the contact pressures in such cases would be much greater than those reported in this study. Another limitation is that the pelvis was fixed and not given any freedom to move as it normally would in vivo—through its articulation with the lumbar spine. This motion influences the load transferred across the hip joint, thus creating a much more complex biomechanical relationship than what was reported in this study. Finally, we do not know how much correction of femoral version can be achieved during HRA. The BHR system was chosen as it continues to show excellent long-term results and is still being used in certain centers around the world [8–10,12,13,35].

It is possible that the ideal component placement to load a hip in extension may not be ideal for a hip loaded in flexion. It is likely that the best performing HRA strikes a compromise that optimizes force transmission and mechanics throughout the functional motion of the joint.

## Conclusion

Reduced femoral version and in extreme cases, femoral retroversion, in HRA influences femoroacetabular impingement and increases joint contact pressure most when the hip is loaded in 90˚ of flexion. Attempting to compensate for this by increasing acetabular cup inclination does decrease the area of impingement but doing so causes a reciprocal increase in joint contact pressure. Ideal component position in HRA needs to consider both the socket and femoral anatomy and should strike a compromise that optimizes force transmission when the hip is loaded throughout the functional motion of the joint. Further research is needed to investigate whether or not these abnormal joint mechanics lead to early implant failure in patients with unrecognizedreduced femoral version. It may be advisable to study femoral neck version preoperatively to better choose hip resurfacing arthroplasty candidates.

## Author Contributions

**Conceptualization:** Jonathan Bourget-Murray, Ashish Taneja, Marwan El-Rich, Samer Adeeb, Kelly Johnston.

**Formal analysis:** Jonathan Bourget-Murray, Ashish Taneja, Sadegh Naserkhaki, Marwan El-Rich, Samer Adeeb, Kelly Johnston.

**Methodology:** Sadegh Naserkhaki, Marwan El-Rich, Samer Adeeb.

**Software:** Sadegh Naserkhaki, Marwan El-Rich, Samer Adeeb.

**Supervision:** James Powell, Kelly Johnston.

**Validation:** Jonathan Bourget-Murray, James Powell, Kelly Johnston.

**Writing – original draft:** Jonathan Bourget-Murray, Kelly Johnston.

**Writing – review & editing:** James Powell, Kelly Johnston.

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
