## [Decision Letter · Decision Letter 0]

6 Jan 2021

PONE-D-20-34959

Computational Modelling of Hip Resurfacing Arthroplasty Investigating the Effect of Femoral Version on Hip Biomechanics

PLOS ONE

Dear Dr. Bourget-Murray,

Thank you for submitting your manuscript to PLOS ONE. After careful consideration, we feel that it has merit but does not fully meet PLOS ONE’s publication criteria as it currently stands. Therefore, we invite you to submit a revised version of the manuscript that addresses the points raised during the review process.

We look forward to receiving your revised manuscript.

Kind regards,

Jose Manuel Garcia Aznar

Academic Editor

PLOS ONE

Journal Requirements:

2. Please include your tables as part of your main manuscript and remove the individual files. Please note that supplementary tables should be uploaded as separate "supporting information" files.

3. Please ensure that you include a title page within your main document. We do appreciate that you have a title page document uploaded as a separate file, however, as per our author guidelines (http://journals.plos.org/plosone/s/submission-guidelines#loc-title-page) we do require this to be part of the manuscript file itself and not uploaded separately.

Reviewers' comments:

Reviewer's Responses to Questions

**Comments to the Author**

1. Is the manuscript technically sound, and do the data support the conclusions?

Reviewer #1: No

2. Has the statistical analysis been performed appropriately and rigorously? 

Reviewer #1: I Don't Know

3. Have the authors made all data underlying the findings in their manuscript fully available?

Reviewer #1: Yes

4. Is the manuscript presented in an intelligible fashion and written in standard English?

Reviewer #1: Yes

5. Review Comments to the Author

Reviewer #1: The aim of this paper is to study the effect of the hip resurfacing position in the mechanical loading scenario manly related with hip contact pressure and hip impingement.

The paper is well write in English and structure is well defined.

The introduction presents a lack of other previous published works related with hip resurfacing offset and position.

The materials and methods presents a lack of information to understand the simulation and the results. Some important technical aspects are critical to analyze the results.

The results are in agreement with previous publications an presents an obvious situation, if the implant position reduces the contact area is obvious increase the contact stress.

Other important and critical aspect is the bone geometry, in the neck of femur and anatomic condition of one patient.

6. PLOS authors have the option to publish the peer review history of their article (what does this mean?). If published, this will include your full peer review and any attached files.

Reviewer #1: No

---

## [Author Response · Author response to Decision Letter 0]

30 Jan 2021

Dear Dr. Anzar, 

RE: PONE-D-20-34959.

 Thank you for reviewing the manuscript of our original research article entitled “Computational Modelling of Hip Resurfacing Arthroplasty Investigating the Effect of Femoral Version on Hip Biomechanics” for consideration for publication in the PLOS One. 

 After careful review of PLOS ONE’s publication criteria, we have edited our manuscript according to the journal’s standards. We hope you find the corrections within our revised manuscript acceptable. Notable changes include:

Level 1 heading used for all major sections. Bold type, 18pt font. 

Level 2 heading used for all sub-sections of major sections. Bold type, 16pt font.

Figure citation corrected across all the text. 

Each figure caption now appears directly after the paragraph in which they are first cited.

All figure titles are now bolded.

Tables are now presented directly after the paragraph in which they are first cited.

Tables are all cell-based.

References with more than six authors now list the first six author names, followed by “et al.”

 The authors would also like to thank the reviewer and academic editor for their comments. We value your insightful input and have attempted to edit our manuscript in a way that reflects these.

Reviewer 1 comment: The introduction presents a lack of other previous published works related with hip resurfacing offset and position.

Author response: There is unfortunately a lack of scientific evidence regarding the impact of femoral retroversion on hip biomechanics and the impact of this hip resurfacing. The purpose of our Finite Element Analysis was to explore this in a computational model given the paucity of research in this field. Insight on femoral retroversion on HRA with respect to impingement and force transmission across the hip joint may encourage further clinical research in this field. 

In light of your comment, we have edited our introduction in order to highlight this issue more clearly. We hope you appreciate the new changes. 

Reviewer 1 comment: The materials and methods presents a lack of information to understand the simulation and the results. Some important technical aspects are critical to analyze the results.

Author response: We have reviewed our Materials and Methods section of our paper. The authors believed to have successfully explained both the computational model techniques used to create the finite element analysis as well as the different models analysed. Did the reviewer feel strongly about any specific aspect of the methods that may need clarity? 

However, we have edited some parts of the methods in order to improve the understanding of our approach. We believe this now reads better and may facilitate understanding. Most of these edits are found in the last two paragraphs of the methods.

The authors wishes to thank both the reviewer and academic editor for their time in reviewing these changes. We hope these are satisfactory to proceed with publication in PLOS One. Should you have further comments of concerns, please feel free to contact us. 

There have been no changes to our financial disclosures. 

Sincerely, 

Jonathan Bourget-Murray, MD CM FRCSC 

Senior Orthopaedic Trauma Fellow 

John Radcliffe Hospital | Oxford University Hospitals NHS Trust 

University of Oxford, Oxford, UK

---

## [Decision Letter · Decision Letter 1]

17 Mar 2021

PONE-D-20-34959R1

Computational Modelling of Hip Resurfacing Arthroplasty Investigating the Effect of Femoral Version on Hip Biomechanics

PLOS ONE

Dear Dr. Bourget-Murray,

Thank you for submitting your manuscript to PLOS ONE. After careful consideration, we feel that it has merit but does not fully meet PLOS ONE’s publication criteria as it currently stands. Therefore, we invite you to submit a revised version of the manuscript that addresses the points raised during the review process.

We look forward to receiving your revised manuscript.

Kind regards,

Jose Manuel Garcia Aznar

Academic Editor

PLOS ONE

Journal Requirements:

Reviewers' comments:

Reviewer's Responses to Questions

**Comments to the Author**

1. If the authors have adequately addressed your comments raised in a previous round of review and you feel that this manuscript is now acceptable for publication, you may indicate that here to bypass the “Comments to the Author” section, enter your conflict of interest statement in the “Confidential to Editor” section, and submit your "Accept" recommendation.

Reviewer #1: (No Response)

Reviewer #2: All comments have been addressed

2. Is the manuscript technically sound, and do the data support the conclusions?

Reviewer #1: Partly

Reviewer #2: Yes

3. Has the statistical analysis been performed appropriately and rigorously? 

Reviewer #1: Yes

Reviewer #2: N/A

4. Have the authors made all data underlying the findings in their manuscript fully available?

Reviewer #1: Yes

Reviewer #2: Yes

5. Is the manuscript presented in an intelligible fashion and written in standard English?

Reviewer #1: Yes

Reviewer #2: Yes

6. Review Comments to the Author

Reviewer #1: Specific comments:

Introduction

The introduction should updated to include some recent studies in Hip resurfacing position, manly the offset and acetabular position.

Uemura, K., Boughton, O.R., Logishetty, K., Halewood, C., Clarke, S.G., Harris, S.J., Sugano, N., Cobb, J.P. A single-use, size-specific, nylon arthroplasty guide: a preliminary study for hip resurfacing (2020) HIP International, 30 (1), pp. 71-77.

Izmin, N.A.N., Todo, M., Abdullah, A.H. Prediction of bone damage formation in resurfacing hip arthroplasty (2019) International Journal of Engineering and Advanced Technology, 9 (1), pp. 5879-5885.

Ramos, A., Soares dos Santos, M.P., Mesnard, M. Predictions of Birmingham hip resurfacing implant offset - In vitro and numerical models (2019) Computer Methods in Biomechanics and Biomedical Engineering, 22 (4), pp. 352-363.

Materials and METHODS

“….usual surgical operation with a layer of cement between the femoral head “

Please identify the thickness of cement mantle in the simulation. Is not an important aspect in the results of contact pressure and MOM.

“…The cortical bone was meshed with 3-node shell elements with a uniform thickness of 1.5mm, …”

Justify why use this option, because the CT scan presents the cortical and cancellous bone? This cortical is important in the model stiffness and can change the results?

“….. The BHR implant and cement were meshed with 8-node hexahedral elements….”

Why the authors use tetrahedral linear elements in the bone and hexahedral elements in the cement?

Why use tetramesh in the bones? and not a solid body if the results are only in the contact between hip resurfacing components

“…., hence the results were insensitive to the mesh size …” what kind of results in this model was insensitive to the mesh size?

To determine the contact area in the hip resurfacing the mesh size is important?

“…..inner surface of the cup with 80μm of clearance….”

Please explain that, because is important to identify the mesh size to guarantee that clearance, 0.08mm.

“… A concentrated load was applied to a reference point defined at the center of the femoral head….”

This point seams the most critical in the study. In addition, is not well defined where the load is applied? In the femoral neck?

“..RV0,..” Please change for neutral

“….This included simulation of hip flexion up to 90° through a virtual maneuver of the femur in the sagittal plane….”

Explain how do you detect the contact?

What is the tolerance? How the model calculate the volume of impingement?

DISCUSSION

The authors do not compare the results of contact stress with other studies.

“…….However, our study does present results of a hip that is loaded in three different positions, which is unparalleled to any previous study.”

This sentence is not true, the published papers presented always 3 direction loads

Reviewer #2: I reviewed a revised version of this paper not being the original reviewer. The paper reads well and to me is clear. Authors seems to have addressed the original reviewer comments which seems to be very minor. In such cases I respect the original review and response and avoid any further comment. However, in this case I would be grateful if authors can add a few lines to the limitations of this paper saying that no direct validation was performed here or somewhere just comment on how confident they are as per validity of the FE results. I understand that the relative comparisons that authors have made remains valuable and perhaps valid.

7. PLOS authors have the option to publish the peer review history of their article (what does this mean?). If published, this will include your full peer review and any attached files.

Reviewer #1: No

Reviewer #2: No

---

## [Author Response · Author response to Decision Letter 1]

16 Apr 2021

Dear Dr. Anzar, 

PONE-D-20-34959R1

 Thank you for reviewing the revised manuscript of our original research article entitled “Computational Modelling of Hip Resurfacing Arthroplasty Investigating the Effect of Femoral Version on Hip Biomechanics” for consideration for publication in the PLOS One. 

 We have carefully reviewed all reviewer comments and addressed all of these. We are convinced this will satisfy Reviewer #1’s comments on our methodology. In fact, there have been notable changes in our Methods sections. Please see below for the list of edits. 

Reviewer #1: Specific comments:

Introduction

We have updated our introduction and have included two recent studies as suggested:

Izmin NAN, Todo M, Abdullah AH. Prediction of bone damage formation in resurfacing hip arthroplasty. International Journal of Engineering and Advanced Technology 2019;9(1): 5879-5885.

Ramos A, Soares dos Santos MP, Mesnard M. Predictions of Birmingham hip resurfacing implant offset - In vitro and numerical models. Comput Methods Biomech Biomed Engin. 2019 Mar;22(4):352-363.

The main changes are found in the second paragraph of the introduction, which now reads:

“In light of these adverse events, several studies have identified risk factors attributed to early failure, which include: implant design, component position and size, female sex, patient age, and surgeon inexperience [8,12,15-17]. As a means to investigate the association between femoral implant malposition on damage formation of the femur, a recent finite element analysis was developed. This model was developed from a 47-year old patient’s computed tomography (CT) image and the loading simulation was that of normal walking condition. The analysis showed that the model experienced the most damage when the femoral head implant was in varus position (>130°), but was reduced significantly when the implant was placed in valgus position (<130°) [16]. On the other hand, Ramos et al. developed experimental and numerical models to analyze whether the positioning of the resurfacing head implant is important in the distribution of bone strains and in the risk of fracture of the femur [17]. They found that valgus position reduces strain distribution in the medial aspect of the femur and brings about a lower shear stress, thus reducing the risk of femoral neck fracture. However, despite appropriate patient selection and precise implant positioning, there continues to be a subset of patients who experience early failure following HRA. Structural abnormalities of the hip may serve as an explanation for some of these failures.”

Materials and METHODS

“….usual surgical operation with a layer of cement between the femoral head “

Please identify the thickness of cement mantle in the simulation. Is not an important aspect in the results of contact pressure and MOM.

The thickness of cement mantle was 0.5 mm. We used one layer of brick solid elements to simulate the cement layer and the minimum size of the elements was 0.5 mm.

“…The cortical bone was meshed with 3-node shell elements with a uniform thickness of 1.5mm, …”

Justify why use this option, because the CT scan presents the cortical and cancellous bone? This cortical is important in the model stiffness and can change the results?

The reviewer is absolutely right about the cortical thickness and its stiffness; however, it does not affect/change results of this research. Thickness of the cortical bone along with its material property determines its stiffness and affects the developed stresses. Since the load was applied as a concentrated force to the head center, there was no developed stresses in the cortical bone. In this research we did not investigated the bone stresses. We only focused on the stresses between the implant head and cup (contact pressure). When reconstructing the cortical bone, its superficial surface was very important for us so we could capture the impingement.

“….. The BHR implant and cement were meshed with 8-node hexahedral elements….”

Why the authors use tetrahedral linear elements in the bone and hexahedral elements in the cement?

Why use tetramesh in the bones? and not a solid body if the results are only in the contact between hip resurfacing components

Our preferred mesh was structured mesh with hexahedral elements technically and computationally. The difference between bone and cement which determined their mesh type was their geometry. The cement was geometrically very regular so we could use structured mesh with hexahedral elements. While cancellous bone had irregular shape and difficult to be meshed using hexahedral elements. 

As the reviewer commented, the bone was not the main focus of this research. If assigning the rigid body we still could perform the analyses and calculate the contact pressure between implant head and cup. In our model both options (rigid body bone or meshed bone) were almost the same in a computational point of view (computational time and size). We used meshed bone model as the preferred one so we could use the same model in future projects were the bone stress is in demand. 

“…., hence the results were insensitive to the mesh size …” what kind of results in this model was insensitive to the mesh size?

To determine the contact area in the hip resurfacing the mesh size is important?

The results were insensitive to the mesh size because we had to use very fine mesh (if the mesh was coarse it definitely could affect the results). In order to mesh the relatively thin implant (mesh it with several layers) and to manage the aspect ratio of the elements, we had to use very small size elements. Finer mesh although affects the results, but negligibly in our case.

“…..inner surface of the cup with 80μm of clearance….”

Please explain that, because is important to identify the mesh size to guarantee that clearance, 0.08mm.

Despite real geometry of other elements of the model, its contact elements were virtual. The clearance of 0.08mm, was determined by property of contact element, not its geometry. In this way, the clearance of the contact became independent from mesh size. We defined the contact by pairing to surfaces. When these two surfaces were getting close together, at one defined distance (which was defined to be 0.08mm in our model) the contact was detected.

“… A concentrated load was applied to a reference point defined at the center of the femoral head….”

This point seams the most critical in the study. In addition, is not well defined where the load is applied? In the femoral neck?

As shown in Fig. 1 (tip of the white arrow), the load was applied as a concentrated force to a single point located on the center of femoral head.

“….This included simulation of hip flexion up to 90° through a virtual maneuver of the femur in the sagittal plane….”

Explain how do you detect the contact?

What is the tolerance? How the model calculate the volume of impingement?

Impingement analysis was performed using Hypermesh (Altair, USA) using virtual maneuver of the femur simulating flexion up to 90°. By this simulation, the angle of flexion at which impingement started was detected and area of impingement was calculated each case. For each case, the flexion rotation was simulated from 1° to 90° at 1 degree rotation steps (the tolerance of rotation). At each step, "trim with surfs/plane" was used. If two surfaces (surface of femoral neck and surface of cup) penetrated each other, then both surfaces would be trimmed; otherwise they would be remained intact. If the surfaces were trimmed, then the trimmed surface and the volume between two surfaces could be selected and area and volume of impingement could be calculated. 

Discussion

We have also updated our discussion as per the reviewer’s suggestions. 

Reviewer #2: I reviewed a revised version of this paper not being the original reviewer. The paper reads well and to me is clear. Authors seems to have addressed the original reviewer comments which seems to be very minor. In such cases I respect the original review and response and avoid any further comment. However, in this case I would be grateful if authors can add a few lines to the limitations of this paper saying that no direct validation was performed here or somewhere just comment on how confident they are as per validity of the FE results. I understand that the relative comparisons that authors have made remains valuable and perhaps valid.

The Authors

---

## [Decision Letter · Decision Letter 2]

17 May 2021

Computational Modelling of Hip Resurfacing Arthroplasty Investigating the Effect of Femoral Version on Hip Biomechanics

PONE-D-20-34959R2

Dear Dr. Bourget-Murray,

We’re pleased to inform you that your manuscript has been judged scientifically suitable for publication and will be formally accepted for publication once it meets all outstanding technical requirements.

Kind regards,

Jose Manuel Garcia Aznar

Academic Editor

PLOS ONE

Additional Editor Comments (optional):

Reviewers' comments:

Reviewer's Responses to Questions

**Comments to the Author**

1. If the authors have adequately addressed your comments raised in a previous round of review and you feel that this manuscript is now acceptable for publication, you may indicate that here to bypass the “Comments to the Author” section, enter your conflict of interest statement in the “Confidential to Editor” section, and submit your "Accept" recommendation.

Reviewer #2: All comments have been addressed

2. Is the manuscript technically sound, and do the data support the conclusions?

Reviewer #2: Yes

3. Has the statistical analysis been performed appropriately and rigorously? 

Reviewer #2: N/A

4. Have the authors made all data underlying the findings in their manuscript fully available?

Reviewer #2: Yes

5. Is the manuscript presented in an intelligible fashion and written in standard English?

Reviewer #2: Yes

6. Review Comments to the Author

Reviewer #2: I have no further comment on this paper, a valuable study for the literature on biomechanics of hip resurfacing.

7. PLOS authors have the option to publish the peer review history of their article (what does this mean?). If published, this will include your full peer review and any attached files.

Reviewer #2: No

---

## [Editor Report · Acceptance letter]

19 May 2021

PONE-D-20-34959R2 

Computational modelling of hip resurfacing arthroplasty investigating the effect of femoral version on hip biomechanics 

Dear Dr. Bourget-Murray:

I'm pleased to inform you that your manuscript has been deemed suitable for publication in PLOS ONE. Congratulations! Your manuscript is now with our production department. 

Kind regards, 

on behalf of

Dr. Jose Manuel Garcia Aznar 

Academic Editor

PLOS ONE